# Chitosan Oligosaccharides Protect Sprague Dawley Rats from Cyclic Heat Stress by Attenuation of Oxidative and Inflammation Stress

**DOI:** 10.3390/ani9121074

**Published:** 2019-12-03

**Authors:** Ruixia Lan, Siqi Li, Qingqing Chang, Zhihui Zhao

**Affiliations:** Department of Animal Science, College of Agriculture, Guangdong Ocean University, NO.1 Haida Road, Mazhang Distinct, Zhanjiang 524-088, China; Lanrx@gdou.edu.cn (R.L.); Lsq_i@outlook.com (S.L.); changqingqing@outlook.com (Q.C.)

**Keywords:** chitosan oligosaccharides, heat stress, antioxidant, anti-inflammatory, Sprague Dawley rats

## Abstract

**Simple Summary:**

Heat stress has negative effects on animal health and performance, and chitosan oligosaccharides (COS) exhibits antioxidant and anti-inflammatory properties. The aim of this study was to evaluate the effects of COS alleviation of oxidative stress and inflammatory response in heat-stressed rats. The results indicated heat stress decreased (*p* < 0.05) growth performance; the relative weight of spleen and kidney; and the level of antioxidant enzymes and IL-10 in liver, spleen, and kidney, while it increased (*p* < 0.05) the MDA and inflammatory cytokines concentration. Dietary COS supplementation enhanced (*p* < 0.05) ADG, the relative weight of spleen and kidney, and the level of antioxidant enzymes and IL-10 in liver, spleen, and kidney. Collectively, COS was beneficial to heat-stressed rats by alleviating oxidative damage and inflammatory response.

**Abstract:**

Chitosan oligosaccharides (COS) exhibits antioxidant and anti-inflammatory properties. The aim of this study was to evaluate the effects of COS on antioxidant system and inflammatory response in heat-stressed rats. A total of 30 male rats were randomly divided to three groups and reared at either 24 °C or 35 °C for 4 h/d for this 7-day experiment: CON, control group with basal diet; HS, heat stress group with basal diet; HSC, heat stress with 200mg/kg COS supplementation. Compared with the CON group, HS significantly decreased (*p* < 0.05) average daily gain (ADG); average daily feed intake (ADFI); the relative weight of spleen and kidney; the level of liver CAT, GSH-Px, T-AOC, and IL-10; spleen SOD, GSH-Px, GSH, and IL-10; and kidney SOD, GSH-Px, T-AOC, and IL-10, while significantly increased the MDA concentration in liver, spleen, and kidney; the liver IL-1β concentration; and spleen and kidney IL-6 and TNF-α concentration. In addition, dietary COS supplementation significantly improved (*p* < 0.05) ADG; the relative weight of spleen and kidney; the level of liver GSH-Px, spleen GSH-Px, GSH, and IL-10; and kidney GSH-Px, while significantly decreased (*p* < 0.05) liver IL-1β concentration under heat stress condition. Collectively, COS was beneficial to heat-stressed rats by alleviating oxidative damage and inflammatory response.

## 1. Introduction

High ambient temperature has negative effects on animal and human health and performance and is responsible for billions of dollars losses to global animal agriculture [1]. Consequently, from a human medicine and agriculture perspectives, researches focusing on the identification and implementation of strategies to improve welfare and performance are essential [2]. Numerous studies have demonstrated that heat stress can induce multiple physiological disturbances, including endocrine disorders [3,4], electrolyte imbalance [5], immune dysfunction [6,7,8], and oxidative stress [9,10,11]. The heat stress response pathway is partly attributed to the increase of pro-inflammatory cytokines and reactive oxygen species (ROS) production [12,13,14]. The increased ROS resulted in stimulated intracellular and extracellular superoxide formation and was responsible for oxidative stress [15]. Heat stress generates ROS, which may modulate the inflammatory transcription factor nuclear factor-κB [16], inducing hormonal and metabolic changes, secretion of inflammatory cytokines [17], and decrease the level of antioxidant enzymes [17]. Former studies demonstrated that the pro-inflammatory cytokines level will increase under heat stress [13,14,18]. In addition, both cytokines and ROS are inflammatory mediators; there is interactive association between oxidative stress and the release of inflammatory cytokines [19,20]. To date, nutritional interventions are widely used to diminish the negative effects of heat stress [2,14,21].

Chitosan oligosaccharides (COS) are the degraded products of chitosan or chitin [22], which is well-known for being soluble in water and less viscous [23], biocompatible [24], non-toxic [25], and mucoadhesive [26]. Innumerable studies have demonstrated that COS exhibits various biology activities including antioxidant [27,28], anti-inflammatory [29,30,31], anti-bacterial [32,33], anti-fungal [34], anti-tumor [35], anti-obesity [36], anti-diabetic [37], calcium absorption enhancing [38], and immune ability enhancing [31,39]. Recent studies also indicated that COS could be used as therapeutic agents in inflammation, suppressing the production of nitric oxide, prostaglandin E_2_, and pro-inflammatory cytokines [39,40,41]. Furthermore, it has been demonstrated that COS can prevent mice from lipopolysaccharide (LPS) challenge by virtue of anti-inflammatory effects as well as antioxidant properties [42]. However, relative few studies were conducted to evaluate the effects of COS on the inflammatory response and antioxidant capacities under heat stress condition. Therefore, the aim of this study to evaluate the effects of COS under cycle heat stress on inflammatory response and antioxidant capacities in Sprague Dawley (SD) rats.

## 2. Materials and Methods

### 2.1. Animal Ethics

The experimental protocol used in the present study was approved by the Animal Care and Use Committee of Guangdong Ocean University, China (SYXK-2018-0147).

### 2.2. Chemicals and Reagents

COS was purchased from Jiangsu Xinrui Biotechnology Co., Ltd. (HPLC purity 95%, deacetylation degree over 95% and average molecular weight below 32kDa). The commercial kits used for the determination of enzymes activities of superoxide dismutase (SOD), catalase (CAT), glutathione peroxidase (GSH-Px), glutathione (GSH), and total antioxidant capacity (T-AOC), the content of malondialdehyde (MDA), enzyme-linked immunosorbent assay (ELISA) kits of interleukin-1β (IL-1β), interleukin-6 (IL-6), interleukin-10 (IL-10), and tumor necrosis factor-α (TNF-α), as well as BCA protein assay kit were purchased from Nanjing Jiancheng Bioengineering Institute (Jiangsu, China). Other chemicals used were analytical grade and purchased from Shanghai chemical agents’ company, China.

### 2.3. Animals, Experiment Design, and Diets

The 30 male Sprague Dawley rats (6–8 weeks, 119.65 ± 3.93 g) used in this study were obtained from Beijing Administration Office of Laboratory Animal (Beijing, China). The rats were individually housed in polycarbonate cages with soft wood granulate floors and kept at 24 °C, with a 12 h light-dark cycle. After a week of acclimatization, 30 rats were randomly divided into one of three groups with 10 rats in each group for this 7-day experiment: (1) CON, control group with basal diet; (2) HS, heat stress group with basal diet; (3) HSC, heat stress with 200 mg/kg COS supplementation. The COS supplementation level was according to our former study (data was not shown). All rats had free access to diets and drinking water. The basal diets were formulated to meet the nutritional requirements recommendation by American Institute of Nutrition-93 diet [43], the composition listed in Table 1. To induce heat stress, the rats in HS and HSC groups were exposed to cyclical heat stress conditions (35 °C from 08:00 to 12:00 and 24 °C from 12:00 to 08:00), while the CON group was maintained at 24 °C for 24 h. Body weight, daily water intake, and feed intake were recorded daily.

### 2.4. Tissue Collection

At the end of the experimental, all rats were fasted for 12 h and 6 rats were randomly selected and euthanized under general anesthesia with diethyl ether. The liver, kidney, and spleen were harvested and weighed, then washed in ice-cold saline solution. All organ weights were expressed as a percentage of live BW (g/kg), based on the studies by Wu et al. [44]. For biochemical assays, a 10% homogenate of the tissue was prepared in PBS (0.01M, pH 7.4) and centrifuged at 3000× *g* for 10 min at 4 °C, and the supernatant was harvested for further assays.

### 2.5. Assay of Antioxidant, Pro-Inflammatory, and Anti-Inflammatory Indices in Plasma and Tissue Samples

The level of MDA, SOD, CAT, GSH-Px, GSH, T-AOC, IL-1β, IL-6, IL-10, and TNF-α in liver, kidney, and spleen were measured according to the manufactures’ protocol.

### 2.6. Statistical Analysis

Data were analyzed by one-way ANOVA using the GLM procedures of SAS (V9.1, SAS Inst., Inc., Cary, NC, USA). Data for the CON group versus the HS group or the HS group versus the HSC group were done to compare the effects of heat stress or COS supplementation y = under heat stress. Significant differences between treatment means were determined by using Duncan’s multiple range tests, and *p* < 0.05 was considered significant.

## 3. Results

### 3.1. Apparent Biological Response

The growth performance and relative organ weight of SD rats are shown in Table 2. There was no significant difference in FCR among treatments. The HS group showed a significant decrease (*p* = 0.0048) in ADFI compared with the CON group. The ADG, the relative weight of spleen and kidney, in the HS group was higher (*p* < 0.05) than that in the CON and HSC group.

The effects of cyclic heat stress on daily water intake are shown in Table 3. The HS group showed a higher (*p* < 0.05) daily water intake than the CON group on days 1, 2, 3, 5, 6, and 7. There was no significant effect on daily water intake in the HSC group compared with HS group, except a decreasing (*p* < 0.05) on day 6.

### 3.2. MDA, Antioxidant Enzymes, GSH, and T-AOC in Liver

The effects of cyclic heat stress on MDA, antioxidant enzymes, GSH, and T-AOC in liver are shown in Figure 1. No significant difference in the activity of SOD (Figure 1B) and GSH (Figure 1E) among treatments. The HS group showed increase (*p* < 0.05) in the concentration of MDA (Figure 1A), as well as decrease (*p* < 0.05) in the activity of CAT (Figure 1C), GSH-Px (Figure 1D), and T-AOC (Figure 1F) compared with the CON group.

### 3.3. MDA, Antioxidant Enzymes, GSH, and T-AOC in Spleen

The effects of cyclic heat stress or with COS supplementation during heat stress on MDA, antioxidant enzymes, GSH, and T-AOC in spleen are shown in Figure 2. There was no significant difference in the activity of CAT (Figure 2C) and T-AOC (Figure 2F) among treatments. The HS group showed increase (*p* < 0.05) in the level of MDA (Figure 2A), as well as decrease (*p* < 0.05) in the activity of SOD (Figure 2B), GSH-Px (Figure 2D), and GSH (Figure 2E) compared with the CON group. The HSC group showed increase (*p* < 0.05) in the activity of GSH-Px and GSH compared with the HS group.

### 3.4. MDA, Antioxidant Enzymes, GSH, and T-AOC in Kidney

The effects of cyclic heat stress during heat stress on MDA, antioxidant enzymes, GSH, and T-AOC in kidney are shown in Figure 3. There was no significant difference in the activity of CAT (Figure 3C) and GSH (Figure 3E) among treatments. The HS group showed increase (*p* < 0.05) in the level of MDA (Figure 3A), as well as a significant decrease in the activity of SOD (Figure 3B), GSH-Px (Figure 3D), and T-AOC (Figure 3F) compared with the CON group. The HSC group showed increase (*p* < 0.05) in the activity of GSH-Px compared with the HS group.

### 3.5. IL-1β, IL-6, IL-10, and TNF-α in Liver

The effects of cyclic heat stress during heat stress on IL-1β, IL-6, IL-10, and TNF-α in liver are shown in Figure 4. There was no significant difference in the level of IL-6 (Figure 4B) and TNF-α (Figure 4D) among treatments. The HS group showed increase (*p* < 0.05) in the level of IL-1β (Figure 4A), as well as decrease in the level of IL-10 (Figure 4C), compared with the CON group. The HSC group showed decrease (*p* < 0.05) in the level of IL-1β compared with the HS group.

### 3.6. IL-1β, IL-6, IL-10, and TNF-α in Spleen

The effects of cyclic heat stress on IL-1β, IL-6, IL-10, and TNF-α in spleen are shown in Figure 5. There was no significant difference in the level of IL-1β (Figure 5A) among treatments. The HS group showed increase (*p* < 0.05) in the level of IL-6 (Figure 5B) and TNF-α (Figure 5D), as well as decrease in the level of IL-10 (Figure 5C) compared with the CON group. The HSC group showed increase (*p* < 0.05) in the level of IL-10 compared with the HS group.

### 3.7. IL-1β, IL-6, IL-10, and TNF-α in Kidney

The effects of cyclic heat stress on IL-1β, IL-6, IL-10, and TNF-α in kidney are shown in Figure 6. There was no significant difference in the level of IL-1β (Figure 6A) among treatments. The HS group showed increase (*p* < 0.05) in the level of IL-6 (Figure 6B) and TNF-α (Figure 6D), as well as decrease in the level of IL-10 (Figure 6C) compared with the CON group.

## 4. Discussion

The current study revealed that dietary COS supplementation improved rat performance, relative weight of spleen and kidney, enhanced antioxidant capacities, and alleviated inflammatory response in rats under heat stress.

In the current study, it is clearly demonstrated that rat average daily gain (ADG) is reduced due to reduction in average daily feed intake (ADFI), which was consistent with those of Azed et al. [15] and Wang et al. [8], who reported depressed growth performance, feed intake, and feed efficiency under heat stress. Furthermore, heat-stressed rat fed COS exhibited a better ADG, which is consistent with a previous study that oligosaccharides improve growth performance in broilers under heat stress [9].

Water is an essential nutrient for animals, especially under heat stress condition. Water intake during heat stress is a limiting factor for survival and performance, due to its fundamental role in temperature regulation and maintenance of hydration balance. Water restriction enhances the detrimental effects of heat stress on animal performance. Under heat stress, the increasing panting and sweating increase water losses, while the water used by metabolism is reduced [45]. The current results indicated that rat drank more water under heat stress for water replenishment.

Heat stress proved to disturb the balance between the production of ROS and the antioxidant systems in animals [15,46]. MDA is the end product of lipid peroxidation, and is the indicator of lipid peroxidation. Former studies have indicated that heat stress can induce lipid peroxidation [9]. Consistently, in the current study, heat-stressed rat exhibited a higher MDA content in liver, spleen, and kidney, whereas dietary COS supplementation numerically reduces MDA content in liver, spleen, and kidney, implying that COS administration could attenuate oxidative damage by heat stress. In addition, SOD, CAT, GSH-Px, GSH, and T-AOC activities were checked to estimate the response of enzymatic and non-enzymatic antioxidant systems, respectively. Compared to the CON group, rats in HS group showed a significant decrease in CAT and GSH-Px activity in liver, a significant decrease in GSH-Px and GSH activity in spleen, and a significant decrease in SOD, GSH-Px, and T-AOC activity in kidney. However, dietary COS supplementation increased the activity of GSH-Px in liver, spleen, and kidney, as well as GSH in spleen, which may alleviate the heat-stress-induced antioxidant damage. Glutathione antioxidant system plays a pivotal role in cellular defense against ROS, GSH-Px function either directly or indirectly as antioxidants, and GSH plays an important role in scavenging reactive oxygen species [47]. It has been reported that dietary chitosan supplementation could enhance the serum SOD, CAT, and GSH-Px of diquat challenged weaning pigs [48]. In addition, dietary chitosan supplementation can prevent lipid peroxidation induced by LPS and retain GSH and CAT activity [42].

Former studies have demonstrated that heat stress significantly up-regulated the level of pro-inflammatory cytokines TNF-α, IL-1β, and IL-6, whereas it down-regulated anti-inflammatory cytokines IL-10 [8,13]. In the current study, the higher levels of IL-1β in liver and IL-6 and TNF-α in spleen and kidney were found in the HS group compared to the CON group, while cytokine IL-10 level was lower. Meanwhile, in the liver, the lower IL-1β but greater IL-10 concentration was observed in the HSC group compared to the HS group, as well as greater IL-10 concentration in spleen, which indicated that COS plays a virtue role in alleviating inflammatory response. Similar to our results, previous studies indicated that dietary COS supplementation reduced pro-inflammatory cytokines IL-1β, IL-6, and TNF-α, as well as increased anti-inflammatory cytokine IL-10 [41,42,49].

## 5. Conclusions

Collectively, our studies indicated that dietary COS supplementation was beneficial to heat-stressed rats, protecting them from oxidative damage and alleviating the inflammatory response.

## Figures and Tables

**Figure 1 animals-09-01074-f001:**
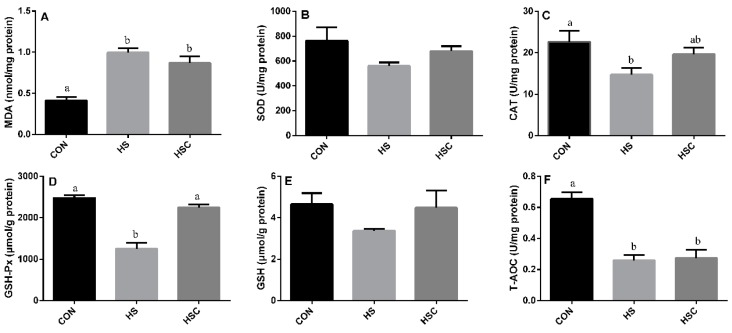
Effects of cycle heat stress or chitosan oligosaccharides (COS) supplementation during cyclic heat stress on MDA, antioxidant enzymes, GSH, and T-AOC in liver. Values are mean ± standard error, n = 6. The values having different superscript letters are different (*p* < 0.05). CON, thermoneutral (24 °C for 24 h/d); HS, heat stress (35 °C for 4 h/d followed by 24 °C for 20 h/d); HSC, heat stress with 200 mg/kg COS supplementation. MDA, malondialdehyde; SOD, superoxide dismutase; CAT, catalase; GSH-Px, glutathione peroxidase; GSH, glutathione; T-AOC, total antioxidant capacity.

**Figure 2 animals-09-01074-f002:**
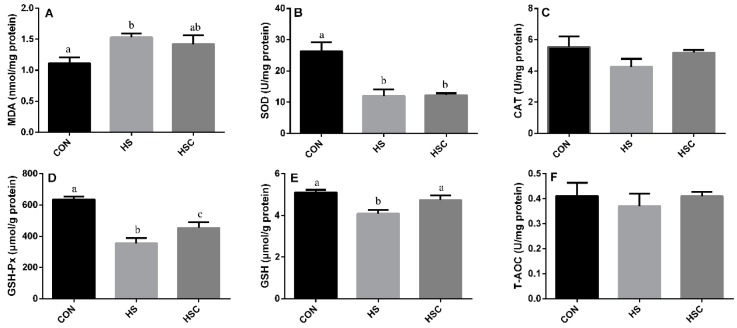
Effects of cycle heat stress or chitosan oligosaccharides (COS) supplementation during cyclic heat stress on MDA, antioxidant enzymes, GSH, and T-AOC in spleen. Values are mean ± standard error, n = 6. The values having different superscript letters are different (*p* < 0.05). CON, thermoneutral (24 °C for 24 h/d); HS, heat stress (35 °C for 4 h/d followed by 24 °C for 20 h/d); HSC, heat stress with 200 mg/kg COS supplementation. MDA, malondialdehyde; SOD, superoxide dismutase; CAT, catalase; GSH-Px, glutathione peroxidase; GSH, glutathione; T-AOC, total antioxidant capacity.

**Figure 3 animals-09-01074-f003:**
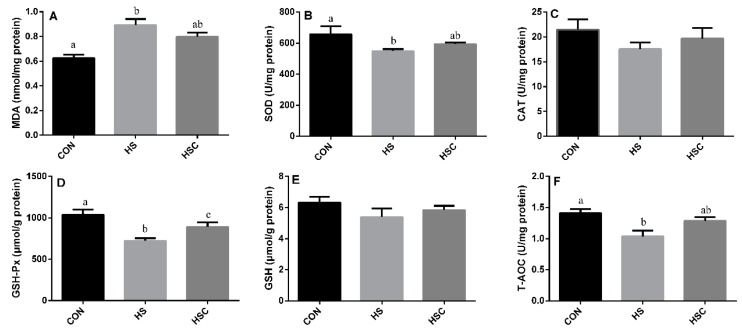
Effects of cycle heat stress or chitosan oligosaccharides (COS) supplementation during cyclic heat stress on MDA, antioxidant enzymes, GSH, and T-AOC in kidney. Values are mean ± standard error, n = 6. The values having different superscript letters are different (*p* < 0.05). CON, thermoneutral (24 °C for 24 h/d); HS, heat stress (35 °C for 4 h/d followed by 24 °C for 20 h/d); HSC, heat stress with 200 mg/kg COS supplementation. MDA, malondialdehyde; SOD, superoxide dismutase; CAT, catalase; GSH-Px, glutathione peroxidase; GSH, glutathione; T-AOC, total antioxidant capacity.

**Figure 4 animals-09-01074-f004:**
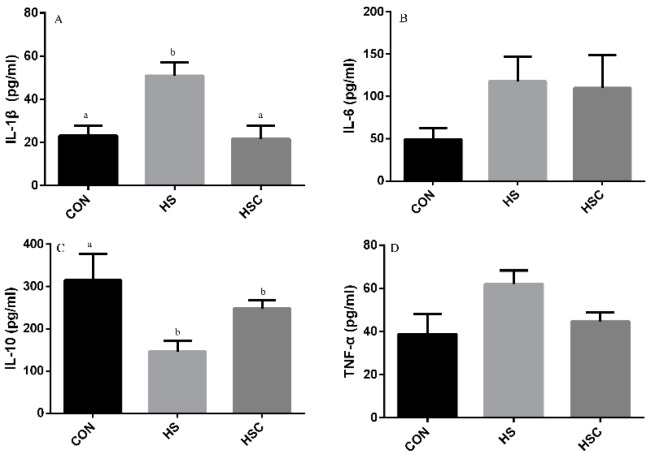
Effects of cycle heat stress or chitosan oligosaccharides (COS) supplementation during cyclic heat stress on IL-1β, IL-6, IL-10, and TNF-α in liver. Values are mean ± standard error, n = 6. The values having different superscript letters are different (*p* < 0.05). CON, thermoneutral (24 °C for 24 h/d); HS, heat stress (35 °C for 4 h/d followed by 24 °C for 20 h/d); HSC, heat stress with 200 mg/kg COS supplementation. IL-1β, interleukin-1β; IL-6, interleukin-6; IL-10, interleukin-10; TNF-α, tumor necrosis factor-α.

**Figure 5 animals-09-01074-f005:**
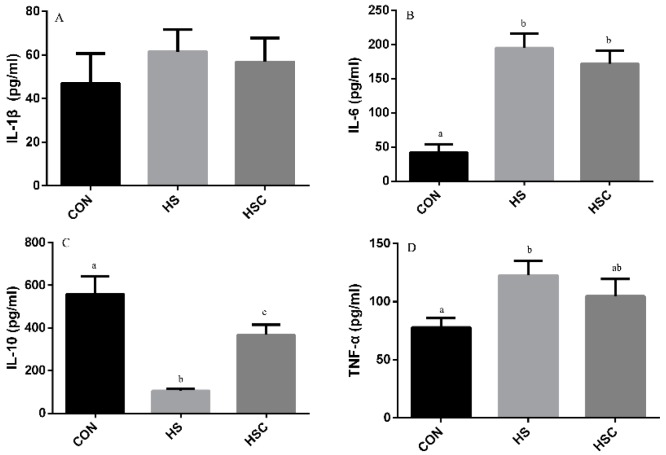
Effects of cycle heat stress or chitosan oligosaccharides (COS) supplementation during cyclic heat stress on IL-1β, IL-6, IL-10, and TNF-α in spleen. Values are mean ± standard error, n = 6. The values having different superscript letters are different (*p* < 0.05). CON, thermoneutral (24 °C for 24 h/d); HS, heat stress (35 °C for 4 h/d followed by 24 °C for 20 h/d); HSC, heat stress with 200 mg/kg COS supplementation. IL-1β, interleukin-1β; IL-6, interleukin-6; IL-10, interleukin-10; TNF-α, tumor necrosis factor-α.

**Figure 6 animals-09-01074-f006:**
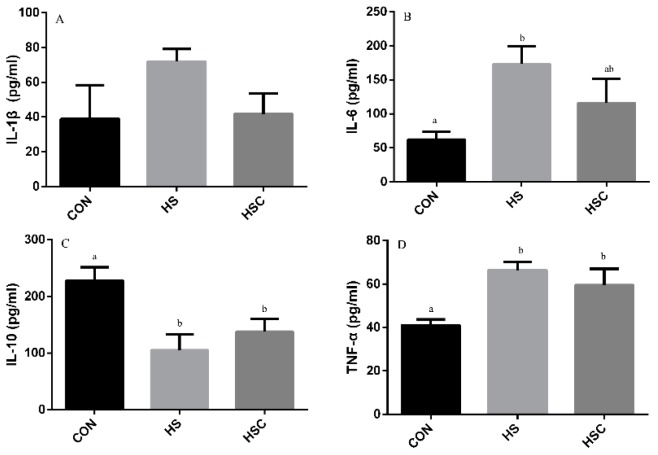
Effects of cycle heat stress or chitosan oligosaccharides (COS) supplementation during cyclic heat stress on IL-1β, IL-6, IL-10, and TNF-α in kidney. Values are mean ± standard error, n = 6. The values having different superscript letters are different (*p* < 0.05). CON, thermoneutral (24 °C for 24 h/d); HS, heat stress (35 °C for 4 h/d followed by 24 °C for 20 h/d); HSC, heat stress with 200 mg/kg COS supplementation. IL-1β, interleukin-1β; IL-6, interleukin-6; IL-10, interleukin-10; TNF-α, tumor necrosis factor-α.

**Table 1 animals-09-01074-t001:** Dietary composition and nutrient content of the basal diet.

Ingredients, g/kg	Basal Diet
Cornstarch	464.0
Casein	140.0
Dextrinized cornstarch	155.0
Sucrose	100.0
Soybean oil	40.0
Cellulose acetate	50.0
Mineral permix ^1^	35.0
Vitamin permix ^2^	10.0
L-Methionine	1.8
L-Cystine	1.8
Choline bitartrate	2.4
Tert-butylhydroquinone	0.01
Gross energy (MJ/kg)	16.22

^1^ Mineral mixture was prepared as AIN-93 (mg/kg of mixture): CaCO_3_, 3.70 × 10^5^; KH_2_PO_4_, 1.96 × 10^5^; K_3_C_6_H_5_O_7_·H_2_O, 7.08 × 10^4^; NaCl, 7.4 × 10^4^; K_2_SO_4_, 4.66 × 10^4^; MgO, 2.4 × 10^4^; FeC_6_H_5_O_7_H_2_O, 6.06 × 10^3^; ZnCO_3_, 1.65 × 10^3^; MnCO_3_, 630; CuCO_3_Cu(OH)_2_H_2_O, 324; NaSiO_3_·9H_2_O, 1.45 × 10^3^; CrK(SO_4_)·12H_2_O, 275; LiCl, 17.4; H_3_BO_3_, 81.5; NaF, 63.5; NiCO_3_·2Ni(OH)_2_·4H_2_O, 30.6; NH_4_VO_3_, 6.6; sucrose was added to make a total of 1 kg. ^2^ Vitamin mixture was prepared as AIN-93 (mg/kg of mixture): Nicotinic, 3.0 × 10^3^; calcium pantothenate, 1.6 × 10^3^; pyridoxine hydrochloride, 700; thiamine hydrochloride, 600; riboflavin, 600; folic acid, 200; D-biotin, 20; cyanocobalamin, 2.5 × 10^3^; a-tocopherol, 1.5 × 10^4^; cholecalciferol, 250; phylloquinone, 75; sucrose was added to make a total of 1 kg.

**Table 2 animals-09-01074-t002:** Effects of cycle heat stress or chitosan oligosaccharide supplementation during cyclic heat stress on growth performance and relative organ weights in Sprague Dawley rats.

Item ^1^	Treatment			*p*-Value	
	CON	HS	HSC	HS ^2^	COS ^3^
Final body weight, g	170.29 ± 10.32	168.74 ± 7.55	169.94 ± 9.15	0.1237	0.1431
ADG, g	3.52 ± 0.38 ^a^	2.43 ± 0.36 ^b^	3.02 ± 0.47 ^a^	0.0013	0.0360
ADFI, g	12.49 ± 1.60 ^a^	10.12 ± 0.58 ^b^	11.19 ± 1.21 ^ab^	0.0048	0.1338
FCR	3.58 ± 0.56	4.24 ± 0.66	3.79 ± 0.74	0.0640	0.1847
Liver (mg g^−1^)	48.86 ± 4.61	43.29 ± 3.11	50.95 ± 10.76	0.1909	0.0824
Spleen (mg g^−1^)	6.27 ± 1.64 ^a^	4.20 ± 0.81 ^b^	5.48 ± 1.60 ^a^	0.0007	0.0144
Kidney (mg g^−1^)	11.29 ± 2.96 ^a^	7.56 ± 1.44 ^b^	9.88 ± 2.87 ^a^	0.0008	0.0148

^1^ Values are mean ± standard error, n = 6. The values having different superscript letters are different (*p* < 0.05). CON, thermoneutral (24 °C for 24 h/d); HS, heat stress (35 °C for 4 h/d followed by 24 °C for 20 h/d); HSC, heat stress with 200 mg/kg COS supplementation. ^2^ CON group vs. HS group. ^3^ HS group vs. HSC group.

**Table 3 animals-09-01074-t003:** Effects of cycle heat stress or chitosan oligosaccharide supplementation during cyclic heat stress on daily water intake in Sprague Dawley rats.

Item/g	Treatment ^1^			*p*-Value	
	CON	HS	HSC	HS ^2^	COS ^3^
Day 1	13.53 ± 2.31 ^b^	19.34 ± 3.40 ^a^	18.75 ± 2.68 ^a^	0.0015	0.6681
Day 2	18.71 ± 3.20 ^b^	30.44 ± 5.35 ^a^	26.77 ± 3.82 ^a^	0.0002	0.0977
Day 3	20.68 ± 3.54 ^b^	25.25 ± 4.44 ^a^	25.13 ± 3.58 ^a^	0.0333	0.9497
Day 4	25.18 ± 4.31	29.58 ± 5.20	28.97 ± 4.13	0.0730	0.7846
Day 5	24.02 ± 4.11 ^a^	34.94 ± 6.15 ^b^	39.15 ± 5.59 ^b^	0.0014	0.1227
Day 6	23.25 ± 3.98 ^a^	30.78 ± 5.41 ^b^	25.51 ± 3.64 ^a^	0.0056	0.0337
Day 7	23.64 ± 4.04 ^a^	32.86 ± 5.78 ^b^	32.33 ± 4.61 ^b^	0.0025	0.8247

^1^ Values are mean ± standard error, n = 6. The values having different superscript letters are different (*p* < 0.05). CON, thermoneutral (24 °C for 24 h/d); HS, heat stress (35 °C for 4 h/d followed by 24 °C for 20 h/d); HSC, heat stress with 200 mg/kg COS supplementation. ^2^ CON group vs. HS group. ^3^ HS group vs. HSC group.

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
