# Peer review of "Chitosan Oligosaccharides Protect Sprague Dawley Rats from Cyclic Heat Stress by Attenuation of Oxidative and Inflammation Stress"

_animals, 2019, doi:10.3390/ani9121074_

Round 1

Reviewer 1 Report

There are still many issues with English and the use of plurals / word choices, this paper needs to be revised by a native or strong English speaker before resubmission for many of these basic errors.

The discussion section states findings from other studies but does not discuss these in relation to the findings of the study presented in any great depth. Rather than just stating comparable findings the authors should discuss similarities and differences.

In Material and Methods, one dietary level of COS has been chosen for this experiment but no justification is provided for this choice.

In terms of editorial quality, the manuscript needs substantial improvements. I have indicated some of them below but the authors need to make a thorough editing revision.

Specific comments

Line 148: Change “The HS group was shown to have a significant decrease” to “The HS group shown a significant decrease” and the others similarity in the full paper.

Line 150-152: “There were no significant effects on daily water intake in the heat-stressed rats with COS supplementation compared with rats exposure to heat stress, except a significantly (P<0.05) decrease on day 6.” This sentences is too long and rephrasing is needed.

Author Response

Comments and Suggestions for Authors

There are still many issues with English and the use of plurals / word choices, this paper needs to be revised by a native or strong English speaker before resubmission for many of these basic errors.

Author response: Thanks for your suggestion. We agree with you, and we ask help from native speaker. Please See Line 11-17, 20-22, 28, 31, 44-53, 59-61, 63-66, 121-124, 131-134, 142-146, 155-160, 169-174, 183-187,196-200, 209-212, 221, 230-234,236-250, 256-261. Thanks a lot.

The discussion section states findings from other studies but does not discuss these in relation to the findings of the study presented in any great depth. Rather than just stating comparable findings the authors should discuss similarities and differences.

Author response: Thanks for your suggestion. We agree with you, and we made changes. 221, 230-234,236-250, 256-261. Thanks a lot.

In Material and Methods, one dietary level of COS has been chosen for this experiment but no justification is provided for this choice.

Author response: Thanks for your suggestion. We choose the dietary level of COS was according to our former study, but data was not shown. Please see line 88. Thanks a lot.

In terms of editorial quality, the manuscript needs substantial improvements. I have indicated some of them below but the authors need to make a thorough editing revision.

Author response: Thanks for your suggestion. We agree with you and made changes. Please See Line 11-17, 20-22, 28, 31, 44-53, 59-61, 63-66, 121-124, 131-134, 142-146, 155-160, 169-174, 183-187,196-200, 209-212, 221, 230-234,236-250, 256-261. Thanks a lot.

Specific comments

Line 148: Change “The HS group was shown to have a significant decrease” to “The HS group shown a significant decrease” and the others similarity in the full paper.

Author response: Thanks for your suggestion. We agree with you, and made changes. Please See Line 121-124, 131-134, 142-146, 155-160, 169-174, 183-187,196-200, 209-212. Thanks a lot.

Line 150-152: “There were no significant effects on daily water intake in the heat-stressed rats with COS supplementation compared with rats exposure to heat stress, except a significantly (P<0.05) decrease on day 6.” This sentences is too long and rephrasing is needed.

Author response: Thanks for your suggestion. We agree with you, and made changes. Please See Line 131-134. Thanks a lot.

Reviewer 2 Report

It was an interesting report of anti-heat stress effects of COS, but followings should be concerned:

1. In the Figures, statistical significance should be apppeared as '* p < 0.05, ** p < 0.01' rather than 'a, b, or ab'.

2. In the Methods, authors should explain how to determine the amount of COS.

3. In the Discussion and Conclusion, authors should duscuss further the role or mechanism of COS. In the manuscript, Discussion seems like a simple repeat of Results rather than a discussion. The anti-lipid peroxidative stress of COS over ROS and anti-inflammatory responses of COS on each organ could be examples.

Author Response

it was an interesting report of anti-heat stress effects of COS, but followings should be concerned:

In the Figures, statistical significance should be apppeared as '* p < 0.05, ** p < 0.01' rather than 'a, b, or ab'.

Author response: Thanks for your suggestion. But we think a, b, or ab is OK, The values having different superscript letters are different (P<0.05). Please see line 127-128, 137-138, 150, 164, 178, 191, 204, and 216. Thanks a lot.

In the Methods, authors should explain how to determine the amount of COS.

Author response: Thanks for your suggestion. In this study, we do not determine the amount of COS in diet, we just use HPLC method to determine its purity. Please see line 72-73. Thanks a lot.

In the Discussion and Conclusion, authors should duscuss further the role or mechanism of COS. In the manuscript, Discussion seems like a simple repeat of Results rather than a discussion. The anti-lipid peroxidative stress of COS over ROS and anti-inflammatory responses of COS on each organ could be examples.

Author response: Thanks for your suggestion. We agree with you and made changes. Please See Line 221, 230-234,236-250, 256-261. Thanks a lot.

Reviewer 3 Report

Although this manuscript provides some interesting scientific results several deficiencies should be addressed before acceptance for publication in the Animals.

If treatment was performed on the last day, the weight of the rat on the last day should also be listed.

Statistical analysis should be done in a more appropriate way.

Please clarify any differences from previous reports.

Food and Chemical Toxicology (2008), 46(2), 710-716.

Author Response

Although this manuscript provides some interesting scientific results several deficiencies should be addressed before acceptance for publication in the Animals.

Author response: Thanks for your affirmation and help.

If treatment was performed on the last day, the weight of the rat on the last day should also be listed.

Author response: Thanks for your suggestion. We have added, please see Table 2, line 126.

Statistical analysis should be done in a more appropriate way.

Author response: Thanks for your suggestion. We have made changes, please see line 114-118, Thanks a lot.

Please clarify any differences from previous reports.

Food and Chemical Toxicology (2008), 46(2), 710-716.

Author response: Thanks for your suggestion. I think this study was quite different from the former study. Thanks a lot.